# Magnetotransport on the nano scale

Philip Willke[1], Thomas Kotzott[1], Thomas Pruschke[2] & Martin Wenderoth[1]

Transport experiments in strong magnetic fields show a variety of fascinating phenomena like the quantum Hall effect, weak localization or the giant magnetoresistance. Often they originate from the atomic-scale structure inaccessible to macroscopic magnetotransport experiments. To connect spatial information with transport properties, various advanced scanning probe methods have been developed. Capable of ultimate spatial resolution, scanning tunnelling potentiometry has been used to determine the resistance of atomic-scale defects such as steps and interfaces. Here we combine this technique with magnetic fields and thus transfer magnetotransport experiments to the atomic scale. Monitoring the local voltage drop in epitaxial graphene, we show how the magnetic field controls the electric field components. We find that scattering processes at localized defects are independent of the strong magnetic field while monolayer and bilayer graphene sheets show a locally varying conductivity and charge carrier concentration differing from the macroscopic average.

[1] IV. Physikalisches Institut – Solids and Nanostructures, University of Goettingen, 37077 Göttingen, Germany. [2] Institut für Theoretische Physik, University of Goettingen, 37077 Göttingen, Germany. Correspondence and requests for materials should be addressed to P.W. (email: philipwillke@gmail.com) or to M.W. (email: martin.wenderoth@uni-goettingen.de).

To elucidate the scattering mechanisms of electrons in a solid, the dependence of the electrical resistance on an external magnetic field, the so-called magnetoresistance (MR), has been a versatile tool connecting theoretical considerations with macroscopic transport measurements[1,2]. The origin of the particular MR is often found on a nanometre scale. Examples are manifold, ranging from the giant MR[3,4], weak localization[1] or simply structural disorder[5–7]. Here the MR reflects the scattering mechanisms induced by atomic-scale defects and nanostructures or the presence of local variations in conductivity and mobility.

Magnetotransport measurements in graphene have been of particular interest since its discovery due to exceptional transport properties including a remarkably high mobility[8,9]. The latter is naturally limited by defects as a source of scattering[5,10–18]. Due to the small spatial extent their influence on transport is often difficult to access. Dissecting different sources of scattering or detecting inhomogeneities in doping or conductivity becomes thus a challenging task. Large-scale transport measurements combined with spatially resolving techniques, such as electron microscopy, helped to disentangle delocalized and localized contributions of electron transport[5,11]. Using scanning tunnelling potentiometry (STP) in previous studies on graphene allowed conclusions on the underlying scattering mechanism at localized defects by the magnitude[12,15] or the position[13] of the voltage drop.

Here we introduce a high magnetic field low-temperature STP set-up to extract the (magneto-)resistance of localized defects. We can show that the resistances of all examined defects are independent of magnetic field strongly differing from pristine sample regions. For monolayer graphene (MLG) and bilayer graphene (BLG) sheets, we find local variations in both conductivity and charge carrier concentration that also differ from the macroscopic mean values of the sample. We are able to derive a consistent picture of magnetotransport down to the atomic scale that could up to now only be discussed by theory[19]. Since none of the transport mechanisms are exclusively dedicated to graphene's unique electronic structure, our findings can be generalized to transport in other systems.

## Results

**Magnetic-field scanning tunnelling potentiometry.** Figure 1a depicts the experimental low-temperature (6 K) STP set-up[13,20]. A transverse magnetic field up to 6 T perpendicular to the current direction can be applied. In a first step the MR of one of our samples (epitaxial graphene on SiC(0001)[13,21]) can be determined macroscopically *in situ* (Fig. 1b) showing mainly a positive quadratic slope $R(B) \propto (\mu B)^2$ with small corrections at low fields due to weak localization[22,23] (for all samples, see Supplementary Fig. 1). In Fig. 1c, we show a typical sample region of MLG and BLG. In our experiment the strong quadratic MR is a consequence of the device geometry[24] $L \sim W$. In combination with the Lorentz-force induced by the magnetic field, electrons get deflected (see Supplementary Note 1) leading to a non-trivial potential drop as demonstrated in Fig. 1d. Here resistor network simulations are shown as a function of magnetic field $B$. These have been obtained by using finite-element method simulations (see Supplementary Notes 1 and 2 and Supplementary Figs 1 and 4). Being below the quantum limit $\mu B \leq 1$, this pronounced MR in Fig. 1b is especially visible in devices with MR geometry, which we particularly chose here to detect small changes in potential and consequently in resistance on a local scale (see Supplementary Note 1 and Supplementary Fig. 3). Whereas the magnetic field dictates the overall shape of the potential drop in the sample on a large scale, local defects and variations in

mobility lead to inhomogeneous voltage drops. This is demonstrated in Fig. 1e for the sample surface area shown in Fig. 1c in the range between −6 and 6 T. As can be seen for the zero-field case the voltage drop is localized at steps, wrinkles and interfaces in contrast to the MLG and BLG sheets[12,13]. For finite magnetic field, the change in direction of the voltage drop agrees with the macroscopic direction.

**Local Hall-effect measurements.** For increasing magnetic field the voltage is also dropping in $y$ direction, visualizing the emerging Hall field on the nano scale. In Fig. 2a,b, we show the spatially averaged voltage drops across the sample area in Fig. 1e in $x$ and $y$ direction, respectively. For the voltage drop in $x$ direction (Fig. 2a), we find it to be monotonous with additional contributions from the local defects. Their influence diminishes with increasing magnetic field. In contrast, the voltage drop in $y$ direction (Fig. 2b) changes in sign and increases with $B$. Moreover, it is inhomogeneous due to the presence of the defects, predominately due to the centred monolayer island in this data set.

Comparing the absolute value of the local potential at the position of the tip as well as the average field components $E_x$ and $E_y$ as a function of $B$ (Fig. 2c–e) to the values of the macroscopic resistor network simulations (see Supplementary Note 1 and Supplementary Figs 1 and 3) allows to quantitatively analyse the Hall field. In the simulations, the experimental macroscopic MR curves shown in Fig. 1b are fitted yielding an average (macroscopic) conductivity $\langle \sigma \rangle$ and charge carrier concentration $\langle n_e \rangle$ (fitted values, see Supplementary Table 1). These are denoted as averages here, since they contain mixed contributions from monolayer and bilayer areas as well as the influence of local defects (for $\langle \sigma \rangle$). The respective potential and electric fields for these averaged $\langle \sigma \rangle$ and $\langle n_e \rangle$ are also shown in Fig. 2c–e. Both experimental data and simulations are in excellent agreement. Consequently, despite the local inhomogeneities, the macroscopic average for $\langle \sigma \rangle$ and $\langle n_e \rangle$ is restored on a scale of $\sim$500 nm, that is, when averaged over a larger scale of defects and single MLG/BLG areas. Note that the electric field and current density components can strongly vary across the sample (see Supplementary Note 2 and Supplementary Fig. 4). It is therefore necessary to know the position of the measurement which we can precisely derive from the measured potential curve $V(B)$ in Fig. 2c (see Supplementary Note 3 and Supplementary Fig. 5).

From the data points in Fig. 2e, we are able to determine the (local) charge carrier concentration $n_e \propto j_x(x, y, B) \cdot B/E_y(x, y, B)$, for which we find $n_e = (1.32 \pm 0.12) \times 10^{13} \, \text{cm}^{-2}$ (evaluation, see Supplementary Note 4).

**Control and monitoring of the electric fields.** The additional influence of the magnetic field can nicely be pronounced by spatially resolved maps of the electric field components $E_x$ and $E_y$ in Fig. 3a,b, respectively. It shows how the current flow around the defect can be controlled by the magnetic field while being read out by STP. For comparison, we added resistor network simulations of the area reduced to its major structural changes, the two MLG areas and a wrinkle on the left. For the magnetic field dependence, we assumed the simplest model including a quadratic change with $B$ for the MLG/BLG sheets and $B$-independent defects (see Supplementary Note 5 and Supplementary Fig. 7).

**Local magnetic field dependence of defects and sheets.** For now, we found that on the nano scale < 500 nm the voltage drop becomes inhomogeneous due to defects and subsequently different current paths lead to large deviations for the electric field

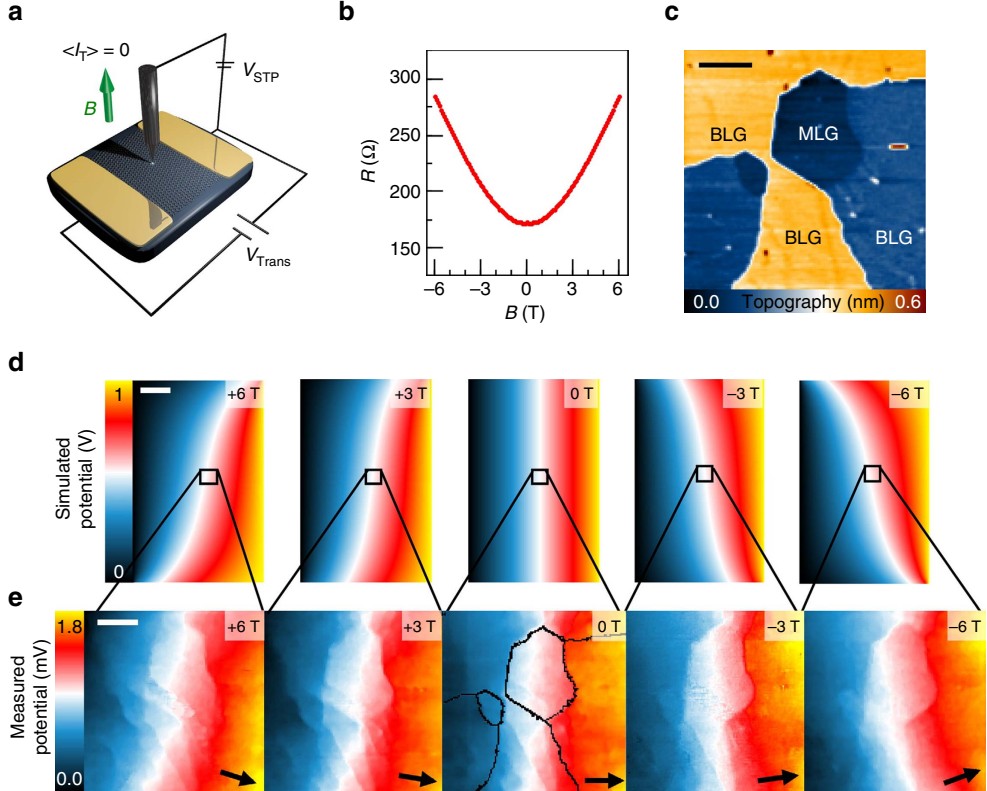

**Figure 1 | Scanning tunnelling potentiometry with applied magnetic field.** (**a**) Schematic of the set-up: large samples of epitaxial-grown graphene on SiC(0001) consisting of MLG and BLG are contacted in two-terminal geometry. The voltage $V_{STP}(x, y)|_{I_T=0}$ necessary to compensate a net tunnel current $I_T$ is recorded and mapped. It represents the voltage drop along the sample induced by the cross voltage $V_{Trans}$. A transverse magnetic field up to 6 T can additionally be applied. (**b**) Macroscopic resistance $R(B)$ of one of the investigated samples as a function of magnetic field $B$. (**c**) Topography of a typical sample area showing a MLG island surrounded by BLG ($440 \times 440 \, nm^2$, scale bar, 100 nm, $I_T = 0.2 \, nA/V_{Bias} = -50 \, mV$). (**d**) Magnetic field-dependent potential landscape for one of the samples in **b** for a cross voltage $V_{Trans} = 1 \, V$ obtained by resistor network simulations ($1.6 \, mm \times 1.1 \, mm$, scale bar, 250 μm, see Supplementary Notes 1 and 2 and Supplementary Figs 1–4). (**e**) Local potential maps for different magnetic fields ($-6 \, T / -3 \, T / 0 \, T / +3 \, T / +6 \, T$) for the sample region in **c** with black lines indicating the steps, arrows the direction of predominate electron flow (scale bar, 100 nm, $V_{Trans} = 4.2 \, V/j_{Macro}(0 \, T) = 15.3 \, A \, m^{-1}$).

in $x$ and $y$ direction. Thus, in the following the magnetic field dependence of the single contributions of graphene MLG/BLG sheets and defects to the resistance is evaluated (Fig. 4). For the topography (Fig. 4a) and potential maps acquired at different $B$ fields (Fig. 4b,c), we show an averaged section in Fig. 4d. Here the electric field $E_x$ increases on the MLG/BLG areas for the 5 T case compared to that at 0 T. Since for a constant current density $j$, this increased electric field $E_x$ corresponds to a higher resistance (higher voltage drop per unit area), this qualitatively reflects the positive quadratic MR found in Fig. 1b. For a quantitative evaluation of the change in $E_x$, an analysis taking into account the exact position on the sample is needed (see Supplementary Notes 2 and 3 and Supplementary Figs 4 and 5). The local sheet conductivity $\sigma = j/E_x$ can be extracted from the electric field $E_x$ for 0 T, its magnetic field dependence gives access to the local charge carrier concentration $n_e$ (detailed discussion on evaluation, see Supplementary Note 6 and Supplementary Figs 8 and 9). Both quantities are shown in Fig. 4e evaluated for a large number of sheets and data sets. We find a large spread of values for both sheet conductivity $\sigma$ and charge carrier concentration $n_e$ up to a factor of 10 indicating local inhomogeneities (s.d. $\Delta\sigma$ and $\Delta n_e$ indicated in Fig. 4e, see also Table 1).

In contrast to the MLG/BLG areas, the voltage drop $\Delta V$ of the two localized defects in Fig. 4d and thus their defect resistances $\rho_{Defect} = \Delta V/j$ remains constant for different $B$ fields. In Fig. 4f, we show the defect resistances as a function of magnetic field for

all extended defects in our epitaxial graphene sample, for example, ML/BL interfaces, wrinkles on BLG and substrate steps. Apparently, for all defect types the defect resistance remains constant effectively leading to a vanishing contribution at high fields (Figs 2a and 4d), since the resistance contribution of the sheets in contrast still increases here (Fig. 1b). Tables 1 and 2 summarize the results for the sheets and the defect resistances, respectively.

## Discussion

The charge carrier concentration $n_e = (1.32 \pm 0.12) \times 10^{13} \, cm^{-2}$ extracted from the local Hall measurements fits perfectly with the macroscopic value for BLG[25] and results from scanning tunnelling spectroscopy (see Supplementary Note 4 and Supplementary Fig. 6). While the latter also allows to extract $n_e$ our method based on local voltage probes keeps the advantage that no a priori knowledge on the electronic structure is needed. Moreover, detection of smaller doping becomes additionally difficult in scanning tunnelling spectroscopy due to the presence of the pseudo-gap for graphene[23]. Despite the good agreement, the Hall field in Fig. 2b clearly shows local deviations manifested in a non-linear voltage drop. This is attributed to variations in mobility and charge carrier density as well as defects, since they determine the local current density resulting in a severely changed Hall field on a scale $< 500 \, nm$.

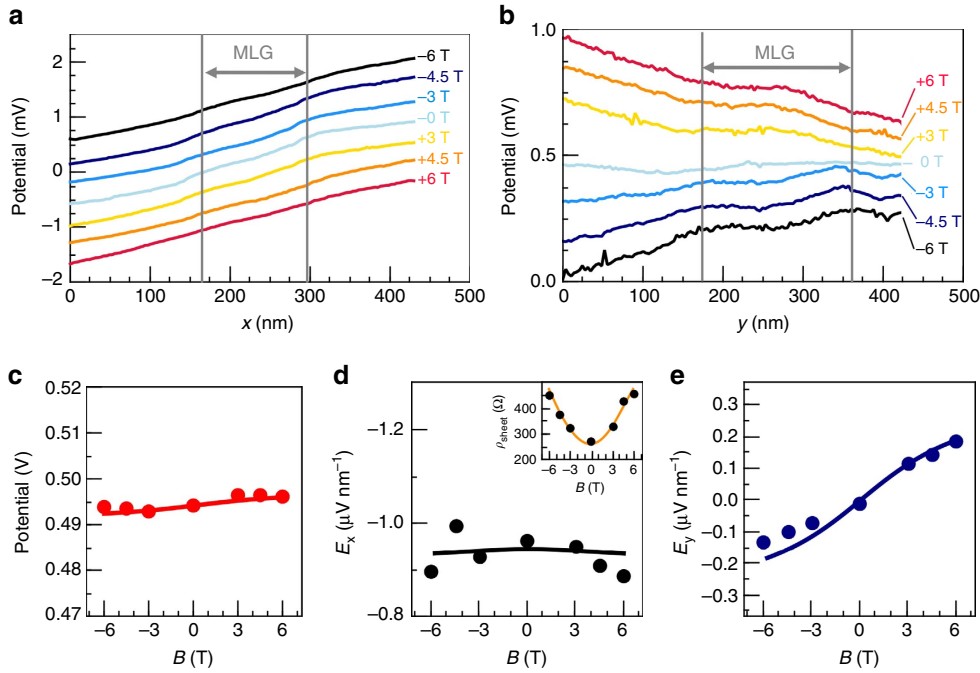

**Figure 2 | Hall measurements on the atomic scale.** (**a**) Potential across the sample region shown in Fig. 1c in x direction (averaged in y direction) and (**b**) in y direction (averaged in x direction). Lines have been shifted relative to each other. (**c**) Average potential as a function of magnetic field. (**d,e**) Electric field components $E_x$ and $E_y$ as a function of magnetic field $B$ derived by linear fits from the data in **a,b**. The lines are the results of the macroscopic finite-element simulation as shown in Fig. 1d at the precise position of the measurement (see Supplementary Note 3 and Supplementary Fig. 5). For comparison, the experimental electric fields are normalized to $V_{Trans} = 1$ V. The inset shows $E_x/j_x$ with the macroscopic sheet resistance (yellow, taken from Fig. 1b).

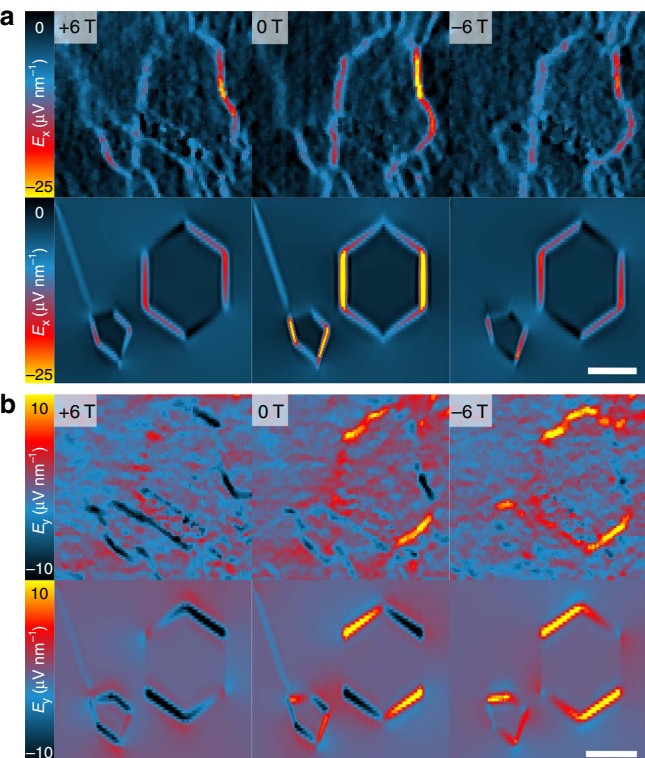

**Figure 3 | Local change of electric fields.** (**a**) Electric field component $E_x$ for the area surrounding the MLG island derived from the potential maps in Fig. 1e ($-6$ T/0 T/ $+6$ T, area: $350 \times 350$ nm$^2$). The lower row shows finite-element method simulations (see Supplementary Note 5) of the area with simple geometries for the ML/BL interfaces and the wrinkle on the left. (**b**) Analysis of the electric field component $E_y$ analogous to **a**.

In addition, the resistor network simulations are also able to reproduce well the changes in electric field components with applied $B$ field on a local scale in Fig. 3, reflecting the change in electron flow around the centre MLG island. Given that this is a classical model neglecting quantum mechanical effects as for example, weak localization[22,23] or Klein tunnelling[26] and only takes into account the main structural features this is quite remarkable. It demonstrates how using the magnetic field the direction of electron flow can be controlled on a nano scale.

For the MLG and BLG sheets, the conductivity $\sigma$ given in Table 1 is higher than macroscopically observed, which obviously stems from the fact that the macroscopic conductivity still contains the influence of steps and interfaces. For the three samples studied in the framework of this work, a decrease in defect concentration showed consequently a higher macroscopic conductivity (Supplementary Fig. 10 and Supplementary Table 1). The MLG sheet resistance shown in Fig. 4e and Table 1 agrees with previous transport measurements using Hall bars[27] and STP measurements[12,13]. The average values of $n_e$ for MLG and BLG are in excellent agreement with spatially averaged values from ARPES[25]. Mobilities are as high as reported for defect-free graphene areas grown under Argon atmosphere[21]. Thus, the transport properties of the ultra-high vacuum (UHV)-grown samples are as good as the highest reported values on SiC when excluding the contribution of the defects. Moreover, the proportional trend $n_e \propto \sigma$ (dashed lines in Fig. 4e) suggests that local variations in $\sigma$ are governed by local variations in $n_e$. This can be caused by the graphene buffer layer as well as stacking faults in BLG[5,28]. Especially the graphene buffer layer can affect both $\sigma$ by local scattering potentials as well as $n_e$ by local changes in doping[29]. (See Supplementary Note 4 and Supplementary Fig. 6). In addition, the large s.d. for both $n_e$ and $\sigma$ suggests that the inhomogeneity of the buffer layer leads to a spread of local resistance. This was previously observed in STP measurements

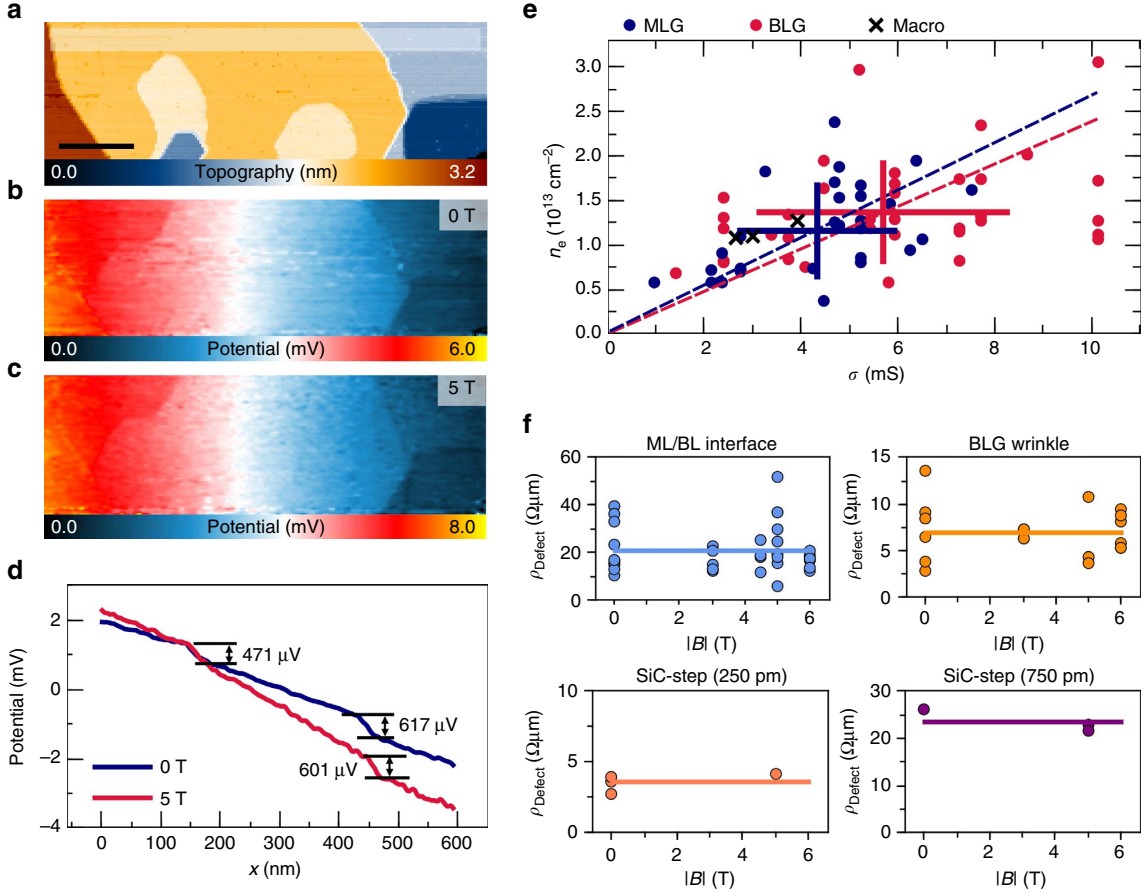

**Figure 4 | Magnetic field dependence of the resistance for graphene and defects.** (**a**) Topography of a microscopic section showing MLG and BLG areas in addition to SiC-substrate steps and ML-BL interfaces (scale bar, 100 nm). (**b**) Corresponding potential map $V_{STP}$ for 0 T and (**c**) for 5 T. Both maps have been rescaled to the same current density $j = 17.7\,Am^{-1}$ ($I_T = 0.3\,nA/V_{Bias} = 50\,mV$). Except for the difference in scale, both maps show a similar voltage drop. (**d**) Sections of the transport map. The voltage drop at the step/interface positions corresponding to the defect resistances is similar for both curves. The slope of the lines corresponding to the sheet resistances show a clear dependence on the magnetic field. (**e**) Conductivity $\sigma$ and charge carrier concentration $n_e$ derived for MLG (blue) and BLG (red) sheets from the change in voltage drop shown in **d** (see Supplementary Note 6). Additionally, the macroscopic values obtained from the MR curves in Fig. 1b are plotted (black, see Supplementary Table 1). The solid lines show the s.d. $\Delta n$ and $\Delta \sigma$ for both MLG and BLG. Dashed lines are a guide to the eye with the slope of the inverse MLG/BLG mobility $(e\mu)^{-1}$. (**f**) Resistances of all localized defects and their change with magnetic field. The lines indicate the $B$-independent average.

**Table 1 | Results for the conductivity σ, charge carrier concentration $n_e$ and mobility μ for macroscopic averaged measurements as well as MLG and BLG with their respective s.d.'s Δσ and $\Delta n_e$.**

|  | σ (Δσ) (mS) | $n_e$ ($\Delta n_e$) ($10^{13}\,cm^{-2}$) | μ ($m^2\,V^{-1}\,s^{-1}$) |
|---|---|---|---|
| Macro | 2.65–3.94 | 1.09–1.25 | 0.155–0.197 |
| MLG | 4.3 ± 0.4 (1.6) | 1.15 ± 0.10 (0.51) | 0.234 ± 0.027 |
| BLG | 5.7 ± 0.5 (2.6) | 1.36 ± 0.10 (0.55) | 0.261 ± 0.027 |

**Table 2 | Results for defect resistance $\rho_{Defect}$ for the different types of defects in SiC-graphene and the change in magnetic field.**

|  | $\rho_{Defect}$ (0 T) (Ωμm) | $\rho_{Defect}$ (>0 T) (Ωμm) | $\rho_{Defect}$ (Ωμm) |
|---|---|---|---|
| ML/BL interface | 22.5 ± 11.7 | 19.7 ± 10.1 | 20.6 ± 7.5 |
| Wrinkle | 7.0 ± 3.2 | 6.7 ± 2.6 | 6.9 ± 2.1 |
| SiC step (250 pm) | 3.4 | 4.1 | 3.6 |
| SiC step (750 pm) | 25.9 | 22.2 | 23.4 |

While the first column shows the values for $B = 0$ T, the second one averaged over all data points with applied magnetic field. The last column yields the total average. No errors for the SiC-substrate steps are given due to the small number of data points.

without magnetic field[17]. The sheet resistance increases by a factor of 2 when going from low temperatures (4 K) to room temperature[21,27] and is almost constant in our samples at low temperatures (<30 K)[23]. Therefore, it is likely that the interaction with the buffer layer is still lowering the conductivity compared to graphene on other substrates[30,31]. The conductivity for BLG is slightly higher than for MLG. It is not simply given by twice the value of MLG, since only one bilayer band is populated at this doping concentration[25]. Additionally, a lower doping in the upper layer and decreasing influence of the buffer layer lead to the

conductivity given in Table 1. (Further discussion on the local conductivity, see Supplementary Note 7).

The constant MR for the localized defects allows to draw conclusions on the underlying scattering mechanisms. A decrease in doping caused by detachment from the substrate present for SiC steps and wrinkles has been previously suggested to explain the voltage drop without magnetic field[14]. This model needs to be

extended, since also a graphene sheet with a different carrier density would show a $B^2$ dependence. Instead a change in doping can be described as a potential barrier from a quantum mechanical point of view. Indeed for the transmission $T$ through a magnetic potential barrier based on wave function matching the MR remains constant, since the wave vector components $k_x/k_y$ barely change for barriers with a small extent (see Supplementary Note 8 and Supplementary Fig. 11). For the ML/BL interface, the scattering due to wave function mismatch[12] and interlayer tunnelling[13] has been discussed as the main contribution in absence of a magnetic field. Also, these scattering mechanisms do not change significantly with magnetic field explaining the same behaviour observed for ML/BL interfaces. Though a variety of magnetic properties of this interface has been discussed including interface states and interface Landau levels[32,33], circulating edge states[34], they do not influence the resistance of this defect. Additionally, an angle-dependent transmission[26,35] inevitably induced by the magnetic field does not play a role for the defects and their resistance.

Combining magnetotransport measurements with scanning probe methods opens a new path to tackle a wide range of transport phenomena on the atomic scale. For studies on a mesoscopic scale, we suggest that this method can easily be implanted in an atomic force microscope set-up using Kelvin probe force microscopy[16]. We here demonstrate the different roles of localized defects and pristine sample areas for the build-up of a classical quadratic MR. In the past, the MR in highly inhomogeneous systems[6,7] including BLG[5] has been investigated intensively leading even to a linear MR in case of sufficient disorder. To test the existing theories[36,37], magnetotransport-STP will be an excellent tool, while the results of this work already demonstrate how different structural contributions change the local and macroscopic magnetic-field dependence. In addition, magnetic tunnel junctions[3,4], quantum Hall physics in graphene[27] as well as weak localization phenomena[10,23] are future candidate systems bearing magnetoresistive effects on the nano scale.

## Methods

**Sample preparation.** Samples with epitaxial MLG and BLG are prepared by thermal decomposition of $n$-doped 6H-SiC(0001)[21] at $T = 1,400$–$1,600\,°C$ under UHV ($10^{-10}$ mbar). The samples (2 mm × 7 mm) are electrically contacted *ex situ* with gold contacts of 100 nm thickness by thermal evaporation using a shadow mask. After reinsertion into the UHV chamber, the samples are heated up to $350\,°C$ for 30 min to eliminate surface contaminations before they are transferred *in situ* to a homebuilt low-temperature scanning tunnelling microscope. All measurements were performed at 6 K sample temperature.

**Scanning probe measurements.** STP measurements are taken at every image point by adjusting the electrochemical potential at the tip at fixed tip-sample distance. For STP, the applied bias voltage is switched off while only the transport potential across the sample remains. The potential at the tip is adjusted in a way that the tunnelling current $I_T = 0$. Subsequently, the voltage $V_{STP}(x, y)|_{I_T = 0}$ necessary to compensate the net tunnel current is recorded (see Fig. 1a). This voltage $V_{STP} = \frac{\mu_{ECP}}{e}$ has been referred to as the local electrochemical potential, which is here inherently defined by the STP method[2,20]. Thermovoltage contributions have been eliminated as described in ref. 13. The measurements are made at different values of the electron current in the sample plane, especially at zero and forward and reversed current as defined by the potential applied to the sample contacts. The details of our specific set-up are published elsewhere[20]. A superconducting coil magnet implemented in the microscope was used to create a strong magnetic field at the position of the sample. Due to the high stability of the system the magnetic field can be changed while staying in tunnelling contact between tip and sample. This allows us to take STP measurements at the same position as a function of the magnetic field.

**Data availability.** The MR curves, potential data sets and values for conductivity, charge carrier concentration and defect resistances are available from the authors.

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

## Acknowledgements

This work was supported by the Deutsche Forschungsgemeinschaft (DFG) priority programme 1459 Graphene. We thank A. Heinrich, K. Pierz and H.B. Weber for fruitful discussions and B. Spicher for expert technical assistance.

## Author contributions

M.W. and P.W. planned the experiments; P.W. and T.K. carried out the experiments and the data analysis. P.W. wrote the manuscript; all authors discussed the results and commented on the manuscript.

## Additional information

**Competing interests:** The authors declare no competing financial interests.

**Publisher's note**: 

