## [Peer Review File · Nature Communications]

Reviewers' Comments:

Reviewer #1 (Remarks to the Author)

The article titled "Magnetotransport on the nano scale" by Dr. Wenderoth et al is a well-written manuscript that reports on the demonstration of low-temperature scanning tunneling potentiometry on epitaxial graphene/6H-SiC under large magnetic fields (up to 6T). The work is both technically sound and scientifically rigorous, backed up by numerous detailed calculations and simulations of the experimentally observed signals. Furthermore, the results are presented with sufficient clarity that other researchers would welcome when attempting to reproduce them. However, I would suggest that the authors include a brief experimental section where they describe in more detail technical aspects of their instrument.

The technical achievement of the presented experimental work is indisputable, however, in my opinion it lacks two crucial ingredients:

1. An expanded introduction where the authors clearly elaborate on the usefulness of their experimental technique in elucidating magnetotransport phenomena in other settings where nanoscale resolution is necessary. This could substantially increase the interest in this article from the solid state physics /materials science community.

2. The reported work, as it stands, lacks a clear demonstration of the power of this new technique. In my opinion, reporting the demonstration of STP under high-magnetic fields in a system that ends up exhibiting nanoscale transport phenomena with very weak modulation due to the magnetic field does not serve to highlight the need nor the impact of this new technique.

I think that in addition to the reported results on graphene, an additional experimental case study on any of the large class of materials that exhibit magnetoresistance would be most welcome.

Addressing the above two points, especially (2), would further strengthen the reported results and increase their impact.

Reviewer #2 (Remarks to the Author)

I congratulate the authors on a very study of the magnetic field dependence of transport in single and double layer graphene, using scanning tunneling potentiometry. The results are strong and convincing, and the paper is well written. I am particularly grateful for the detailed information available in the supplementary material. Without this information, the main manuscript would not only be much harder to understand, but also impossible to verify.

I recommend publication of this paper in its present form.

Reviewer #3 (Remarks to the Author)

The manuscript reports on the local measurements of the electric potential inside an epitaxial graphene sheet in the presence of magnetic field. Combining the scanning tunneling potentiometry with the magnetotransport measurements, the authors discuss the role of atomic-scale defects (steps, wrinkles, and interfaces between the monolayer and bilayer parts) in the magnetoresistance. A comparison with numerical modelling based on the so-called resistor network simulations is presented. The authors claim that the local voltage measurements show that the scattering off localized defects is independent of magnetic field, whereas the bulk resistivity increases with increasing magnetic field and is argued to reveal inhomogeneities of the charge concentration (and hence of the local conductivity). These statements are not unexpected, but the experimental approach is rather novel. The measurements performed are of interest to the large community of graphene researchers, as well as to a broader community belonging to the

field of transport phenomena at the nanoscale. The presented experimental results are of a high quality.

However, the interpretation of the experimental findings and the corresponding theoretical part of the manuscript presented in Supplemental Information appear to be not quite convincing.

1. The resistor network model employed for the modelling is not properly specified in Sec. I of Supplemental Information (SI). In particular, it is not clear from the manuscript how this model is formulated, what the parameters justifying the use of this model are, how the randomness is introduced and so on. Since the major claims of the manuscript are based on the comparison with simulations within this model, the authors should clearly describe the approach used in their simulations.

2. The authors claim that the parabolic magnetoresistance (MR) "stems from the increased path of an electron in the presence of magnetic field". They also write "for the magnetic field dependence we assumed the simplest model including a quadratic change with B for the MLG/BLG sheets". Furthermore, below Eq. (1) in SI they write that "the quadratic change in RESISTANCE with magnetic field" is "included" there. However, if one inverts the matrix in Eq. (1), the diagonal components of the resulting resistivity matrix will be independent of B. Apparently, the authors believe that they could simply take $1/\sigma_{xx}$ from this matrix without inverting the matrix as a whole, in order to get ρ_{xx} . This is, of course, wrong:
$$\rho_{xx} = \sigma_{xx} / (\sigma_{xx}^2 + \sigma_{xy}^2) \neq 1/\sigma_{xx}.$$
Therefore, the appearance of the quadratic MR is not properly explained in this work. Given the network model is also not explained (see above), I do not see the origin of the parabolic MR in the system under study. Before going into subtle details, the authors should convincingly explain the main effect which is by no means obvious.

3. In connection with the above, readers would highly benefited from an additional panel in Fig. 1 showing, in addition to Fig. 1b, the macroscopic Hall resistivity. It would also be desirable to perform the measurements at different temperatures, which might help in identification of the mechanism of the parabolic MR (it is important whether it temperature independent), and different degrees of (possible controlled) disorder.

4. The very notion of the local conductivity is only then well defined, when the variation of conductivity on the scale of a mean free path is small, otherwise the inhomogeneities should be considered microscopically. Therefore, it is desirable to compare the average transport mean-free path obtained from the macroscopic resistivity with the typical spatial scales of the carrier-density inhomogeneities. Furthermore, it would be useful to estimate the equilibration length due to electron-electron collisions and compare it with the scale of inhomogeneities. It is only when this equilibration length is shorter than this scale, the local electrochemical potential is well defined and one can use locally the macroscopic description in terms of conductivities.

5. It seems that the size of (atomic-scale) defects discussed in the manuscript is much smaller than any of the relaxation lengths. In view of the above, it is totally unclear why one can use an effective medium description to account for such defects.

6. The notation ρ_{Defect} in Fig. 4 and below is very misleading, as ρ is conventionally used for the resistivity per unit area, whereas extended defects are characterized by their resistance (usually denoted by capital R). The beginning of the figure caption in Fig. 4 is unclear: there can be no "Magnetic field dependence of graphene and defects" (graphene is obviously independent of magnetic field) -- the authors probably meant the magnetic field dependence of the corresponding contributions to the resistance.

7. The use of the proportionality sign in the expressions for ρ below Fig. 4 is very unfortunate, as

it suggests that the resistances do depend on the electric fields, as if the authors considered the nonlinear transport regime of high voltages, with the current being a nonlinear function of voltages. In fact, since the r.h.s. of these expressions is divided by the current that itself is proportional to electric fields, the resistances are not proportional to the values of electric field.

8. The concluding part of the manuscript requires serious improvement, as currently it does not state clearly why the main claims of the manuscript are really nontrivial and hence have required such an elaborated study. The only conclusion explicitly presented in the concluding part ("Thus, additional B-dependent scattering cannot...") does not convince me that the manuscript deserves publication in Nature Communications where manuscripts should be important for a wide community of researchers. In its present form, the concluding remarks suggest that the manuscript is more suitable for a specialized journal (the absence of a clear summary of main results only enhances this impression). Moreover, this single conclusion is followed by a disclaimer that the main finding of the manuscript ignores a variety of important phenomena. As such, the authors themselves seem not to be fully convinced by their own claims -- why should a reader be convinced?

To summarize, the experimental part of the manuscript deserves publication in some form, although additional measurements would be very helpful. However, the interpretation and the theoretical part should be substantially improved. I cannot recommend the publication of this manuscript now: major revisions are necessary.

Reviewers' comments:

Reviewer #1 (Remarks to the Author):

The article titled "Magnetotransport on the nano scale" by Dr. Wenderoth et al is a well-written manuscript that reports on the demonstration of low-temperature scanning tunneling potentiometry on epitaxial graphene/6H-SiC under large magnetic fields (up to 6T). The work is both technically sound and scientifically rigorous, backed up by numerous detailed calculations and simulations of the experimentally observed signals. Furthermore, the results are presented with sufficient clarity that other researchers would welcome when attempting to reproduce them. However, I would suggest that the authors include a brief experimental section where they describe in more detail technical aspects of their instrument.

We thank the reviewer for this suggestion. A detailed description of the experimental setup is now included in the methods section.

The technical achievement of the presented experimental work is indisputable, however, in my opinion it lacks two crucial ingredients:

1. An expanded introduction where the authors clearly elaborate on the usefulness of their experimental technique in elucidating magnetotransport phenomena in other settings where nanoscale resolution is necessary. This could substantially increase the interest in this article from the solid state physics /materials science community.

We agree with the reviewer that a clear outlook on the applications of our technique is beneficial. We extended the discussion on that issue in a proper introduction along with the required changes for the style of Nature Communications.

2. The reported work, as it stands, lacks a clear demonstration of the power of this new

technique. In my opinion, reporting the demonstration of STP under high-magnetic fields in a system that ends up exhibiting nanoscale transport phenomena with very weak modulation due to the magnetic field does not serve to highlight the need nor the impact of this new technique.

I think that in addition to the reported results on graphene, an additional experimental case study on any of the large class of materials that exhibit magnetoresistance would be most welcome.

Certainly the reviewer is right that the investigation of other systems is highly interesting with this new technique. We feel that a second material system would however go beyond the framework of this paper, especially since defects in graphene are already a system of high interest. As it was mentioned above by the reviewer, the technique itself and the interpretation are not trivial and also a main focus of this paper. In addition, the other focus of this paper, the magnetoresistance of graphene on SiC, already contains in our opinion strong MR-effects as the reviewer asks for. The strong MR (50%) is stemming from the graphene sheet areas along with the geometry of the sample and using STP can be dissected from the defect resistance. In this way, the investigated system contains both areas that change their resistance under the influence of a magnetic field (sheets) and those who do not (defects). Due to the quite general underlying scattering mechanisms (electron-electron/phonon-scattering on the sheets, quantum tunneling at the defects) this behavior should be applicable to a wide range of materials as mentioned in the manuscript.

We believe that another system would either fall into this category or would require a different story (for instance a magnetic tunneling barrier) that draws attention away from the main messages of the paper.

Addressing the above two points, especially (2), would further strengthen the reported results and increase their impact.

Reviewer #2 (Remarks to the Author):

I congratulate the authors on a very study of the magnetic field dependence of transport in single and double layer graphene, using scanning tunneling potentiometry. The results are strong and convincing, and the paper is well written. I am particularly grateful for the detailed information available in the supplementary material. Without this information, the main manuscript would not only be much harder to understand, but also impossible to verify.

I recommend publication of this paper in its present form.

We thank reviewer #2 for her/his recommendation and support to publish our results.

Reviewer #3 (Remarks to the Author):

The manuscript reports on the local measurements of the electric potential inside an epitaxial graphene sheet in the presence of magnetic field. Combining the scanning tunneling potentiometry with the magnetotransport measurements, the authors discuss the role of atomic-scale defects (steps, wrinkles, and interfaces between the monolayer and bilayer parts) in the magnetoresistance. A comparison with numerical modelling based

on the so-called resistor network simulations is presented. The authors claim that the local voltage measurements show that the scattering off localized defects is independent of magnetic field, whereas the bulk resistivity increases with increasing magnetic field and is argued to reveal inhomogeneities of the charge concentration (and hence of the local conductivity). These statements are not unexpected, but the experimental approach is rather novel. The measurements performed are of interest to the large community of graphene researchers, as well as to a broader community belonging to the field of transport phenomena at the nanoscale. The presented experimental results are of a high quality.

However, the interpretation of the experimental findings and the corresponding theoretical part of the manuscript presented in Supplemental Information appear to be not quite convincing.

1. The resistor network model employed for the modelling is not properly specified in Sec. I of Supplemental Information (SI). In particular, it is not clear from the manuscript how this model is formulated, what the parameters justifying the use of this model are, how the randomness is introduced and so on. Since the major claims of the manuscript are based on the comparison with simulations within this model, the authors should clearly describe the approach used in their simulations.

We thank the reviewer for this critique and extended the supplementary information accordingly. Supplementary Note I now contains a detailed description of the resistor-network-simulation along with additional simulations (Supplementary Fig. 1 and 2) that help to understand the modeling. Since a lot of the simulation is automated in the software program COMSOL, we did not describe it in detail, but agree with the reviewer that the reproducibility of our work and the understanding of the main results will certainly benefit from an extended description of this part.

2. The authors claim that the parabolic magnetoresistance (MR) "stems from the increased path of an electron in the presence of magnetic field". They also write "for the magnetic field dependence we assumed the simplest model including a quadratic change with B for the MLG/BLG sheets". Furthermore, below Eq. (1) in SI they write that "the quadratic change in RESISTANCE with magnetic field" is "included" there. However, if one inverts the matrix in Eq. (1), the diagonal components of the resulting resistivity matrix will be independent of B. Apparently, the authors believe that they could simply take $1/\sigma_{xx}$ from this matrix without inverting the matrix as a whole, in order to get ρ_{xx} . This is, of course, wrong: $\rho_{xx} = \sigma_{xx} / (\sigma_{xx}^2 + \sigma_{xy}^2) \neq 1/\sigma_{xx}$. Therefore, the appearance of the quadratic MR is not properly explained in this work. Given the network model is also not explained (see above), I do not see the origin of the parabolic MR in the system under study. Before going into subtle details, the authors should convincingly explain the main effect which is by no means obvious.

We agree with the reviewer that the phrasing we used was confusing. The reviewer is of course right and we did not believe by any means that $\rho_{xx} = 1/\sigma_{xx}$. We changed the paragraphs and we also feel that the extended supplementary information helps to eliminate these confusions.

- We rephrased the part “stems from the increased path of an electron in the presence of magnetic field” in the manuscript and extended the description in both the manuscript as well as Supplementary Note I.
- In Supplementary Note 1(a) we now derive explicitly the conductivity tensor from the elementary Drude equations including the Lorentz-Force.
- In Supplementary Note 1(c) and Supplementary Fig. 2 we provide additional simulations of the MR with varying sample geometry. Thus, we show how different magnitudes of the quadratic MR arise from the combination of the conductivity/resistivity matrix, the sample geometry and its boundary conditions (contacts).

3. In connection with the above, readers would highly benefited from an additional panel in Fig. 1 showing, in addition to Fig. 1b, the macroscopic Hall resistivity. It would also be desirable to perform the measurements at different temperatures, which might help in identification of the mechanism of the parabolic MR (it is important whether it temperature independent), and different degrees of (possible controlled) disorder.

In a MR geometry the respective Hall field is position dependent and thus a Hall resistivity is not well-defined (See Supplementary Note II and also Supplementary Ref. 4 [Isenberg et al., 1948]). However, the local field in E_y -direction is shown in Fig. 2e (for the position from Fig. 1) for both experiment as well as finite element simulation. This is proportional to the Hall-field E_H (See Supplementary Note IV).

Concerning the identification of the mechanism, we follow the suggestion of the reviewer and treat this issue now in detail in the discussion part of the manuscript.

The (macroscopic) evaluation of the resistance as a function of temperature in the STM is limited by our instrumentation not allowing for a variation in temperature.

However, we discuss previous studies on the transport on SiC-graphene (Emtsev et al. 2009, Jobst et al. 2010, Jobst et al. 2012) including studies on our own samples (Willke et al. Nano Letters 2015). This contains the temperature-dependence of the resistance and also the role of disorder both suggested by the reviewer. For the latter, large scale AFM images are now shown in Supplementary Figure 9 for all samples used in this study to show how different defect densities affect the conductivity and charge carrier concentrations.

4. The very notion of the local conductivity is only then well defined, when the variation of conductivity on the scale of a mean free path is small, otherwise the inhomogeneities should be considered microscopically. Therefore, it is desirable to compare the average transport mean-free path obtained from the macroscopic resistivity with the typical spatial scales of the carrier-density inhomogeneities. Furthermore, it would be useful to estimate the equilibration length due to electron-electron collisions and compare it with the scale of inhomogeneities. It is only when this equilibration length is shorter than this scale, the local electrochemical potential is well defined and one can use locally the macroscopic description in terms of conductivities.

We thank the reviewer for this remark. We included a discussion of the length scales in Supplementary Note VII: *Discussion of local sheet conductivity and defect resistance.*

5. It seems that the size of (atomic-scale) defects discussed in the manuscript is much smaller than any of the relaxation lengths. In view of the above, it is totally unclear why

one can use an effective medium description to account for such defects.

We believe that this refers to the macroscopic average of the electric fields that we derive from the macroscopic MR (Fig. 1b) and that we compare to the local values in 2d,e. We understand the point of the reviewer and agree that describing it as an effective medium would be difficult. We added explaining sentences at the part of the manuscript we believe the reviewer is referring to.

In the text we did not use the term 'effective medium', but rather 'averaged macroscopic resistance' indicating that, if the defect density and the MLG/BLG-ratio stays constant across the sample, an average 'sheet' resistivity may be defined that contains the inhomogeneity including an average contribution of defects. This has been reached for the topography in Fig. 1c (thus on a scale >500 nm, approximately) as shown by the comparison of the electric fields in Fig. 2d,e.

6. The notation ρ_{Defect} in Fig. 4 and below is very misleading, as ρ is conventionally used for the resistivity per unit area, whereas extended defects are characterized by their resistance (usually denoted by capital R). The beginning of the figure caption in Fig. 4 is unclear: there can be no "Magnetic field dependence of graphene and defects" (graphene is obviously independent of magnetic field) -- the authors probably meant the magnetic field dependence of the corresponding contributions to the resistance.

Concerning the notation the reviewer points out a controversy that in our opinion is depending on how one views the properties of a defect. We agree, that a definition "per unit area" does not make sense for 1D defects, but instead a definition that is "independent of geometric properties". In previous works both ρ [*Science* **336**, 1143–1146 (2012), *APL* **105**, 143109 (2014)] and R [*PRL*. **108**, 096601 (2012)] were used. Thus, we prefer here to use " ρ " for a defect resistance to stay consistent with our previous works (*Nat. Commun.* **6**, 6399 (2015), *Carbon* **102** (2016) 470-476).

Moreover, we corrected the caption according to the reviewers suggestion.

7. The use of the proportionality sign in the expressions for ρ below Fig. 4 is very unfortunate, as it suggests that the resistances do depend on the electric fields, as if the authors considered the nonlinear transport regime of high voltages, with the current being a nonlinear function of voltages. In fact, since the r.h.s. of these expressions is divided by the current that itself is proportional to electric fields, the resistances are not proportional to the values of electric field.

We agree with the reviewer that this statement needs clarification and we extended the paragraph accordingly. We here aimed for a hand-waving intuitive argument why the electric field changes. In fact the situation is much more complicated, since the local electric field is strongly depending on the position as discussed in Fig. 2 and the supplement.

We extended the statement with a more precise explanation and also refer to the more detailed description in the supplement.

8. The concluding part of the manuscript requires serious improvement, as currently it does not state clearly why the main claims of the manuscript are really nontrivial and hence have required such an elaborated study. The only conclusion explicitly presented in the concluding part ("Thus, additional B-dependent scattering cannot...") does not convince

me that the manuscript deserves publication in Nature Communications where manuscripts should be important for a wide community of researchers. In its present form, the concluding remarks suggest that the manuscript is more suitable for a specialized journal (the absence of a clear summary of main results only enhances this impression). Moreover, this single conclusion is followed by a disclaimer that the main finding of the manuscript ignores a variety of important phenomena. As such, the authors themselves seem not to be fully convinced by their own claims -- why should a reader be convinced?

We took this critique of the reviewer very seriously and clarified and extended the discussion. We agree that this has been rather short due to the length restrictions by nature physics (Since this is a transfer manuscript) resulting in only a brief discussion. We elaborated the discussion and the summary of our main findings.

To summarize, the experimental part of the manuscript deserves publication in some form, although additional measurements would be very helpful. However, the interpretation and the theoretical part should be substantially improved. I cannot recommend the publication of this manuscript now: major revisions are necessary.

We appreciate the detailed critique of the reviewer. We feel that we have answered all his questions which, in our opinion, lead indeed to an improvement of the manuscript.

Reviewers' Comments:

Reviewer #1 (Remarks to the Author)

The authors have fully addressed all of my comments and made the appropriate changes to the manuscript. I recommend their manuscript for publication.

Reviewer #3 (Remarks to the Author)

The authors have done a serious work improving the manuscript according to referee's comments. My only remaining concern is about the clarification of the origin of parabolic MR.

The authors write in Supplementary Note I(c): "While the physics of MR is captured by Eq. (3)...". This still sounds misleading, as Eq. (3), in fact, captures the physics of NO MAGNETORESISTANCE, as the authors nicely showed in Eq. (2), where the diagonal components ρ_{00} of the resistance tensor are independent of B (no magnetoresistance). This sentence should be, therefore, rephrased.

The authors further argue that the parabolic dependence of MR is due to geometric effects ($W \sim L$) and refer at this point to the discussion in Ref. (3) of Supplemental Information. However, it would be desirable to explain the physical origin of the parabolic MR in more detail already in Supplementary Note I(c), extending the sentences "... and thus the equipotential lines in their vicinity are heavily bent for applied magnetic field." (WHY DOES IT HAPPEN?) "These regions are responsible for the observed MR" (HOW DO THESE REGIONS GIVE RISE TO A PARABOLIC MR?) A couple of sentences written in simple terms (and maybe a simple formula) here would be highly beneficial for readers who are interested in understanding the physics of the phenomenon, and would nicely complement the reasoning based on the computer simulations. This would definitely make the manuscript more self-contained.

With this modification, the manuscript can be published.

Reviewer #3 (Remarks to the Author):

The authors have done a serious work improving the manuscript according to referee's comments. My only remaining concern is about the clarification of the origin of parabolic MR.

- The authors write in Supplementary Note I(c): "While the physics of MR is captured by Eq. (3)...". This still sounds misleading, as Eq. (3), in fact, captures the physics of NO MAGNETORESISTANCE, as the authors nicely showed in Eq. (2), where the diagonal components ρ_0 of the resistance tensor are independent of B (no magnetoresistance). This sentence should be, therefore, rephrased.

We followed this suggestion of the reviewer and rephrased in the first sentences in Supplementary Note 1c.

- The authors further argue that the parabolic dependence of MR is due to geometric effects ($W \sim L$) and refer at this point to the discussion in Ref. (3) of Supplemental Information. However, it would be desirable to explain the physical origin of the parabolic MR in more detail already in Supplementary Note I(c), extending the sentences "... and thus the equipotential lines in their vicinity are heavily bent for applied magnetic field." (WHY DOES IT HAPPEN?) "These regions are responsible for the observed MR" (HOW DO THESE REGIONS GIVE RISE TO A PARABOLIC MR?) A couple of sentences written in simple terms (and maybe a simple formula) here would be highly beneficial for readers who are interested in understanding the physics of the phenomenon, and would nicely complement the reasoning based on the computer simulations. This would definitely make the manuscript more self-contained.

With this modification, the manuscript can be published.

We changed Supplementary Note 1c and extended the section by additional explanatory sentences, why and how the change in potential is affecting the MR. We additionally added the example of a Corbino disk geometry for which the effect of a positive quadratic MR is easier to understand independent of the sample geometry.

Reviewers' Comments:

Reviewer #3 (Remarks to the Author)

I am satisfied with the authors' response to my previous report and by corresponding changes made to the manuscript.
The manuscript can now be published as it is.